# When Pregnancy Coincides with Positive Diagnosis of HIV: Accounts of the Process of Acceptance of Self and Motherhood among Women in South Africa

**DOI:** 10.3390/ijerph182413006

**Published:** 2021-12-09

**Authors:** Sphiwe Madiba

**Affiliations:** Department of Public Health, School of Health Care Sciences, Sefako Makgatho Health Sciences University, Pretoria 0001, South Africa; sphiwe.madiba@smu.ac.za; Tel.: +27-12-521-3093

**Keywords:** antenatal, coping, experience, HIV seropositive, PMTCT, pregnancy, women, South Africa

## Abstract

Literature has highlighted the unique period of vulnerability following an HIV diagnosis during pregnancy. Despite the high burden of HIV among pregnant women in South Africa, the experiences of women diagnosed with HIV during pregnancy have rarely been explored in isolation from those diagnosed at different times. This paper explored the experiences of women who were diagnosed with HIV when pregnant and assessed their emotional recovery beyond diagnosis. The study used a qualitative descriptive phenomenological approach to conduct interviews with women recruited from ART clinics in a health district in South Africa. Participants included 19 women sampled purposively. The interviews were transcribed verbatim and analysed following the thematic approach. Testing positive during pregnancy and being free of symptoms increased the shock, disbelief, and strong emotions exhibited. For the women, the diagnosis of HIV coincided with pregnancy and transformed pregnancy from excitement to anxiety. Although the transition from being HIV negative to becoming HIV positive and pregnant was overwhelming, with the passage of time, the women transitioned to feelings of acceptance. However, the process of acceptance was slow and varied, with some experiencing non-acceptance for extended periods. Non-acceptance of HIV diagnosis has serious adverse public health consequences for the individual. Integrating continuous HIV counselling and culturally appropriate psychosocial care into practice could foster acceptance for pregnant women with HIV diagnosis.

## 1. Introduction

HIV is a public health issue of great importance that continues to disproportionally affect developing countries [1]. The 2020 Joint United Nations Programme on HIV/AIDS (UNAIDS) estimates show that 37.7 million people globally were living with HIV and nearly 53% of all people living with HIV (PLHI) were women and girls [2]. In South Africa, the high burden of HIV persists despite advances in treatment options and increased access to care. Young people, particularly adolescent girls and young women (AGYM), are at the centre of the pandemic [3]. A population-based household survey conducted in 2017 showed that an estimated 7.9 million people, about 14% of South Africans of all ages, were living with HIV. Similar to the global estimates, women of childbearing age have the highest prevalence of HIV infection in South Africa [4]. The 2017 National Antenatal Sentinel HIV Survey data indicated an overall HIV prevalence 30.7% among pregnant women in the country [5].

Globally, HIV testing is the cornerstone of HIV prevention. In many African settings, routine HIV testing and counselling during pregnancy have been associated with an increase in the numbers of women diagnosed with HIV in antenatal care (ANC) [6,7]. ANC provides a distinct opportunity for HIV testing and counselling as fundamental components of prevention of mother-to-child transmission (PMTCT) of HIV programmes [7]. Other critical components of the PMTCT services are linkage to care through the initiation and use of antiretroviral treatment (ART) and postnatal care of the mother and her infant [8]. The uptake of the PMTCT programme in many countries in sub-Saharan Africa (SSA) has significantly increased the uptake of HIV testing services and early diagnosis of HIV among pregnant women [9]. Consequently, for many AGYW pregnancy is often the point at which an HIV diagnosis is made [10,11].

Although HIV has transitioned from an acute fatal disease to a chronic manageable condition through the advances and widespread availability of ART [12,13,14], receiving an HIV diagnosis can be an emotionally challenging time [15,16]. The conceptualisation of HIV as a social and physical death persists in many societies [17]. This is particularly true in developing countries with a high burden of HIV and a high degree of perceived stigma [18,19]. There is indication from studies conducted in developed and developing countries that an HIV positive diagnosis received within the context of routine screening tests can be devastating, can bring about uncertainty, and may be associated with severe emotional distress [14,17,20]. Research suggests that, compared to women who test positive regardless of pregnancy, the response to testing positive during pregnancy is characterised by much more suffering and much stronger an emotional impact on the women [21].

A meta-synthesis found that pregnant women diagnosed with HIV experience initial shock at the diagnosis, intense loneliness, hope, and limited social support [22]. Women who test positive for HIV during routine ANC screening without symptoms are less emotionally prepared to accept the test result [17,20]. Women often experience psychological distress from being pregnant and living with HIV at the same time, distress that takes a toll on their physical, mental, and social functioning [23]. Having to deal with pregnancy, a positive HIV diagnosis, fear of vertical transmission of HIV to the baby, initiating lifelong ART at once, stigma, and the need to disclose the HIV status to significant others make them vulnerable to extremely distressing emotions [14,15,24,25]. The psychological and psychosocial effect of an HIV diagnosis can contribute to poor self-management and delayed engagement in care [14]. For pregnant women with positive test results, the consequences of delayed engagement in HIV care and uptake of ART have implications for the elimination of vertical transmission of HIV.

In spite of the increasing numbers of women testing HIV positive in pregnancy, few studies have investigated the distinctiveness of the experiences of testing HIV positive in pregnancy from the perspectives of women living with HIV (WLHIV) [16,26]. Even fewer studies have examined the role of the women as active participants in the recovery process [15]. Although South Africa and other settings in SSA have a high burden of HIV, how women experience and cope with an HIV diagnosis during pregnancy is not well-understood [14], particularly in the era of increased access to ART and quality of life of mothers after a diagnosis of HIV.

Literature has highlighted the unique period of vulnerability following an HIV diagnosis during pregnancy [25]. However, limited research has been conducted with women to examine the emotional and psychosocial impact after an HIV-positive diagnosis during pregnancy and related coping strategies [14]. Furthermore, most of the limited data are from studies that were conducted in developed countries. The lack of data on how women experience and cope with an HIV diagnosis during pregnancy have implications on the development of public health interventions to increase uptake of ART and retention in care. Initiation of ART as early as possible in the course of new HIV infection can reduce maternal viral load and the risk of vertical transmission in newly acquired maternal HIV infection [27].

To address the existing gap in research, this paper explored the experiences of WLHIV who were diagnosed with HIV when pregnant and assessed their emotional recovery in the context of their own daily circumstances beyond diagnosis. The experiences of women diagnosed with HIV during pregnancy have rarely been explored in isolation from mothers diagnosed at other times [25]. Therefore, understanding the experiences of receiving an HIV diagnosis following antenatal screening is crucial to providing appropriate care and support to foster the wellbeing of the pregnant woman and her unborn child. It is also important to reflect on the impact of receiving an unexpected positive result and the journey to recovery to inform culturally appropriate public health interventions [17].

## 2. Materials and Methods

### 2.1. Study Design and Setting

The data for this paper are based on re-analysis of a bigger study that used descriptive phenomenology to explore and understand how mothers living with HIV experienced mothering a child who acquired HIV through vertical transmission [28]. Descriptive phenomenology was used to understand and describe how the mothers experience the phenomenon of raising a child infected with HIV [29]. Biological mothers living with HIV were selected using purposeful sampling for in-depth study of the mothering phenomenon [30]. Consistent with the tradition of purposeful sampling, only biological mothers were selected because they could offer a meaningful understanding of the phenomenon of interest [31]. The focus of the current paper was only on those women who tested positive during pregnancy.

The study site was 12 primary health care facilities in a sub-district at Tshwane, in Gauteng Province, South Africa. Tshwane’s estimated HIV prevalence among adults is 11.7%, while among pregnant women using antenatal services is estimated at 23.4% [32]. All the health facilities provide PMTCT programmes and offer counselling, testing, and initiation of ART services to adults and children. The health facilities provide services to clients from urban, peri-urban, and informal settlements in the sub-district.

The participants in the study consisted of mothers living with HIV and raising at least one HIV positive child aged between 1–13 years. The mothers were recruited from ART clinics in the sub-district. Mothers with an HIV diagnosis are referred to the mainstream ART clinics from the PMTCT programme when the baby is 18 months old. They were individually selected via purposeful sampling to participate in the study. The nurse clinicians identified those who met the inclusion criteria and referred them to the research team after they had completed their routine consultation for ART refill. The sample for the main study consisted of 28 mothers living with HIV, was influenced by data saturation, and its ability to provide rich data to understand the mothering phenomenon [31]. The sample for the current paper was 19 mothers who received their HIV diagnosis during pregnancy. While the sample was homogeneous, variability and diversity were achieved by selecting mothers with different socio-demographics such as age, marital status, educational status, employment status, and child age [31].

### 2.2. Data Collection

One-on-one interviews were conducted between November 2015 and January 2016, using a self-developed interview schedule with open-ended questions and possible probes. The interview schedule was based on extensive reading from the literature on the subject of mothering and taking into consideration the objectives of the study [33,34]. It was developed in English and later translated from English into the local language (Setswana) and all interviews were conducted in the local language to allow the participants to express themselves freely in their own language. Consistent with the tradition of phenomenology, the participants were asked broad, open-ended questions such as how they experience testing HIV positive in pregnancy and the process of acceptance of the HIV diagnosis. Probes and follow-up questions were asked to elicit meaning from the responses given.

The participants were interviewed in consulting rooms in the health facilities for the sake of privacy. The administration of the informed consent process entailed explaining the purpose of the study, voluntary participation, confidentiality, privacy, and their right to withdraw from the study at any stage. All the participants provided a signed consent form prior to the interviews. Each interview was about 45 min long and was recorded with the permission of the participants.

Prior to the implementation of the study, the research team consisting of a master’s student and research assistants were trained by the author to familiarise them with the interview schedule, the objectives of the study, and informed consent forms. The research team was experienced in conducting in-depth interviews but was briefed on the sensitive nature of the topic.

### 2.3. Data Analysis

Analysis for the main study began with the transcribing of the interviews verbatim in Setswana by the research team who later translated them into English, and checked the transcripts for accuracy against the audio recordings. Data analysis was inductive and followed the thematic approach in line with the tradition of descriptive phenomenology [35]. The data were then re-analysed by the author to coincide with the aim and objectives of the current paper. The same process of using thematic analysis was followed. Analysis began with repeated reading of the transcripts to search and extract statements for meanings to uncover emergent themes and develop a coding matrix. NVivo version 12 [36], a qualitative data analysis package, was utilised for the analysis process. Lastly, the emergent themes were integrated and synthesised into a meaningful whole that captured the phenomenon as experienced by the participants. Quotes from the interviews were used to describe the lived experiences of the participants.

In phenomenology, rigour or trustworthiness is ensured through the thoroughness and completeness of the collection and analysis of the data [37]. To establish rigour, the interviews were in the local language and recorded to enhance verbatim transcription, which is a necessary step to ensure that meaning is not lost and that the data reflect the phenomenon as experienced by the participants. In addition, obtaining interview notes, peer debriefing, analysing the data using NVivo qualitative software, and keeping an audit trail throughout the research process were the key strategies used to establish confirmability in the main study [38]. In the re-analysis of data, rigour was enhanced by immersing in the data and employing the code-recode strategy [39]. The author adopted a reflective attitude throughout the data analysis to ensure that interpretation was free of bias [37].

### 2.4. Ethical Considerations

The study was granted ethical approval from Sefako Makgatho Health Sciences University Research and Ethics Committee (SMUREC/H/214/2015: PG). The Tshwane Health District granted permission to access the health facilities. All the participants provided written informed consent and participation was voluntary. The participants assumed pseudo-names, which were used during the analysis but deleted in this report. The services of a resident social worker were secured to provide counselling for participants who might break down during the interviews; however, none of the participants needed counselling because of the interviews.

## 3. Results

### 3.1. Description of Study Participants

The sample consisted of 19 mothers living with HIV who received an HIV positive diagnosis in pregnancy and were raising a child living with HIV. Their ages ranged from 27–40 years, all were receiving lifelong ART, 13 were single, 13 were unemployed and depended on the child support grant, and 16 had high school education. The ages of their children ranged from 6–11 years with a mean age of 8.5 years. The children were diagnosed with HIV at birth and after birth.

### 3.2. Themes 

Six main themes emerged from the analysis of the interviews, namely the unanticipated HIV diagnosis, fear of premature death, extreme emotional distress, disclosure of HIV diagnosis, acceptance of self and HIV status over time, and life changes after acceptance. Under each theme, a number of sub-themes emerged (Table 1).

#### 3.2.1. The Unanticipated HIV Diagnosis

Most of the participants expressed their disbelief at the discovery of their diagnosis of HIV. Some described feelings of denial and struggling to accept that they were HIV-positive. Most of them were shocked to hear of their HIV infection and disagreed with their HIV-positive status because they had not expected such a diagnosis.


*I did not believe. I kept on asking myself how I got HIV because I was not that kind of person who is sleeping around with different guys, so how did this happen? (Mother of a 6-year-old).*



*I did not believe that my result could be HIV positive. My partner and I have dated since we were still young and I never dated somebody else without him. That is why I have asked myself how I could be positive (Mother of a 6-year-old).*



*It took some time; I was asking myself a lot of questions, that I am positive while I was not running around or changing partners. I was faithful to him and I still have this thing while I was at home all the time, waiting for him. So this is really hurting (Mother of an 8-year-old).*


#### 3.2.2. Fear of Premature Death

Upon learning about their HIV diagnosis, the participants wondered whether they would survive to bring up the unborn child. Most described a deep sense of fear of prematurely dying and leaving their children. They felt like they were going to die and that death was inevitable. 


*Ooh! I was shocked telling myself that I am going to die soon. I saw myself as a person who is dying at any time, it was not easy at all; it was hectic. I felt like I am losing my mind a bit. People were dying then (Mother of a 9-year-old).*



*What came through my mind was death, I thought that I was going to die…, I am going to leave my children behind (Mother of a 6-year-old).*


#### 3.2.3. Extreme Emotional Distress

For most of the participants, pregnancy created an additional level of anxiety. On diagnosis of HIV, the participants recalled what could be described as extreme emotional distress. They were distressed about the possibility that their unborn child could become infected with HIV.


*What makes it difficult is the fact that you become scared of other people and you do not accept the status that you are in, and so it becomes painful all the time. (Mother of an 8-year-old).*



*I felt so much pain I even lost weight. I still have this pain that comes to my mind. I can’t even sleep well at night (Mother of a 6-year-old).*



*It is painful because you ask yourself whether you will live like this your whole life (Mother of a 10-year-old).*


They were overwhelmed by feelings of pain when they learned about the child’s HIV diagnosis. They reported experiencing the same fear of dying, similar to the one they had when they got their HIV positive test results. Their narratives revealed that they perceived a mild cough as an indicator of the imminent death of their children.


*It is not possible for a mother not to worry about dying; it crosses your mind when he gets sick and you wonder is it because he is HIV positive (Mother of a 6-year-old).*



*I worry about the child too, who is going to die first the child or me. This is my concern (Mother of an 8-year-old).*


##### Despair

The participants thought about their diagnosis of HIV a lot and a deep sense of sadness and grief extended into a depressed state in the weeks and months following their diagnosis. They considered the positive HIV diagnosis to be the finish line and feelings of depression and loss of self-worth were common. Some were driven to desperate acts such as feeling suicidal.


*It felt like I had lost a lot and had no life, feels like you can’t continue with anything. I had plans for my life but it felt like I was no longer living (Mother of a 6-year-old).*



*Ahhhh (sigh)…, let me tell you…, I was so stressed…; I did not know what to think. I even felt like killing myself (Mother of a 7-year-old).*



*Sometimes I felt like defaulting treatment and let myself die (Mother of a 10-year-old).*


##### Self-Neglect and Depression

When the participants talked about the weeks, months, or years that followed their HIV infection diagnosis, it became evident that other longer-term responses developed. Along with deep sadness and long-term emotional effects came the lack of motivation, loss of interest in their daily activities, poor eating, isolation from family and friends, and loneliness.


*It took me maybe six months. I even slept at the hospital. It drained me a lot. I got tired emotionally (Mother of a 6-year-old).*



*I did not bath, I did not see the importance of waking up I was feeling down and it was not nice for me (Mother of an 8-year-old).*



*I lost a lot of weight, I kept asking myself if whether I will die and leave my children..., I was stressed I did not talk to anyone, I felt like no one understood my situation (Mother of an 8-year-old).*


##### Guilt, Shame, and Self-Blame

Nearly all the participants reported feelings of guilt and shame. They felt a sense of taking responsibility for the consequences of their actions by blaming themselves. For some, the diagnosis of HIV was mixed with questions about their moral condition such as engaging in unsafe sexual behaviours.


*I had multiple partners before, some of them I was no longer when I tested; so I did not have a clue of where I got this disease. I only blame my past…, why I did not insist on using a condom, I was very ignorant. I blamed myself big time (Mother of a 9-year-old).*



*I feel useless and irresponsible (Mother of a 9-year-old).*



*I should not put blame on someone else because I am the one who risked my life. First thing, when you have sex you must use condom, you have no right to blame someone else (Mother of a 10-year-old).*


The participants were angry with themselves for getting infected but also blamed their partners for the infection. They were of the view that their partners had not been honest with them about their HIV serostatus or the risk of transmitting HIV.


*I felt betrayed, because I was not sleeping around and my partner was the one sleeping around and I had no idea what he was doing outside. So I felt that he betrayed me after I found out I was HIV positive. He confessed that he was the one who infected me with the HIV. I felt much betrayed (Mother of a 9-year-old).*



*I got if from him, my husband is all over the place. I was angry with him, I do not want to lie, and I did not speak to him for a long time (Mother of a 10-year-old).*


While most were angry with themselves for the trust that they had put in their partners, some were angry with the person they believed had infected them.


*I was blaming him big time, I was so angry I was swearing at him, I was shouting, swearing at my man (Mother of a 6-year-old).*


#### 3.2.4. Disclosure of HIV Diagnosis

Disclosure of own HIV diagnosis and that of the child for most of the participants was delayed. The data further revealed that disclosure was selective, as the participants only disclosed to people that could be trusted to keep their HIV diagnosis and that of their child’s secret as. Disclosure was also to people they believed would give them support.

##### Delayed Disclosure

The participants’ narratives revealed that the mothers kept the child’s HIV diagnosis secret from the child and significant others. They used secrecy as a means to protect themselves and their children from stigma and discrimination.


*Eish! Disclosure is a difficult thing because you will tell people, then after disclosure they start to discriminate your child and not allow their children to play with him (Mother of a 7-year-old).*



*I am scared that once they know about my child’s status they would tell their parents and in that way, it would lead to stigma. They would reject her and make fun of her when they play (Mother of a 9-year-old).*


However, the participants were more open about disclosing their HIV diagnosis to significant people in their lives.


*I told my mother and my little sister (Mother of a 6-year-old).*



*I told my friends that I am positive (Mother of a 9-year-old).*



*I talk openly about my HIV status (Mother of an 8-year-old).*



*I told my oldest child, my mom, my siblings, and some relatives (Mother of a 7-year-old).*


##### Outcome of Disclosure

The participants who disclosed their own HIV diagnosis reported both negative and positive reactions. A few were rejected by their families and sexual partners.


*When I disclosed my HIV status to my mother, she told me that she doesn’t want an HIV positive person in her home. You won’t believe when I say that most of the family members don’t want to touch my child because he is HIV positive (Mother of an 8-year-old).*


In contrast, most of the participants received support from the people close to them after disclosure.


*My eldest child offers support and encourages us [his mother and HIV positive sibling] in taking the treatment. My partner sends me money. My brother and younger sisters offer me support (Mother of a 7-year-old).*



*My mother supported me throughout this illness. So, I share everything with her (Mother of a 6-year-old).*


#### 3.2.5. Acceptance of Self and HIV Status over Time

The participants reported that over time they became accepting of their diagnosis and viewed HIV differently. Their narratives indicated that viewing HIV as an infection that affects many people assisted them to get to self-acceptance.


*I told myself that this is just a virus and it can’t kill me, it’s just an illness my body and if I live in a right way daily (healthy life-style and take treatment) with the belief I will live (Mother of a 10-year-old).*



*I told myself that I am not the first person to get these results. I accepted and told myself that if I am not accepting this, I will worry and this will make my immune system to be weak (Mother of an 8-year-old).*


##### Acceptance Is a Process

The process of acceptance of one’s own HIV status was slow as most participants reported being in denial after learning of their diagnosis. Most came to terms with their HIV diagnosis and said that they eventually accepted the diagnosis as a lifelong condition. However, they described periods of non-acceptance were as ranging from months to years.


*You know to accept is hard and very painful. At first, it was very hard I did not believe I was HIV positive…, it took me almost a year to accept. As you go for counselling, they tell you that it is not the end of the world for you. Firstly, you have to take your pills, take care of yourself and then you will live a normal life like everyone (Mother of a 10-year-old).*



*At the end I realized that this is my life I have to accept even though it was not easy but I have tried..., now when I look back I am saying wow I did it (Mother of a 9-year-old).*



*It took me a while to accept. Sometimes when I walk the streets, I feel like people can tell that I am HIV positive or when someone looks at me, I feel like he/she can see me (Mother of an 8-year-old).*


##### Counselling Gives Hope

The participants discussed the importance of receiving continuous counselling, interacting with other PLHIV, and participating in support groups. They believed that receiving counselling was crucial in the acceptance of their status. More importantly, they acknowledged that the counselling they received gave them hope to live.


*To tell the truth the thing that helped me a lot is counselling; the counsellors help (Mother of a 10-year-old).*



*I went through counselling then I became strong. They explained to me that I won’t die. I’m going to get better and be like others who work for themselves, that’s when I accepted. I would have died but the counselling helped me a lot. I thought of suicide, but the counselling helped me a lot (Mother of an 8-year-old).*



*Eish, it is not that easy to accept that you are positive and that you are going to take medication the rest of your life. I decided to join a support group, so that is where I started to accept myself (Mother of a 9-year-old).*


##### Trust in ART

The presence of effective anti-retroviral medication has made it possible for mothers living with HIV and their children to live normal lives. Another way through which participants coped with their diagnosis of HIV was the hope they had in ART.


*I realized that drinking my treatment would help me live long because if I default I will kill myself (Mother of a 10-year-old).*



*Through counselling, I live normally and I am like other people. I have accepted and I am living a normal life. When I don’t take my pills, I feel as though something is missing (Mother of an 8-year-old).*



*I was told that if I take my medication I’ll be alright and live long (Mother of an 8-year-old).*


##### The Importance of Support from Friends and Families

The participants’ narratives indicated that with time they realized the importance of friends and families. Those who talked about their illness with their family had received emotional support to cope with the situation related to their HIV diagnosis. 


*What keeps me going is that my mother supports me with everything. My mother gives me support like nobody’s business. It is like no matter what happens she supports me (Mother of a 7-year-old).*



*I accepted and took it as though it is another illness because my sister explained to me. I was lucky to find someone who was willing to listen, I realised that I could live long if I took care of myself (Mother of an 8-year-old).*



*My children also support me but my aunt support me more than them (Mother of a 7-year-old).*


#### 3.2.6. Life Changes after Acceptance

Most participants had to change their lifestyle as part of their acceptance of the HIV diagnosis and motherhood. They mentioned feeling motivated to care for themselves to support their children or family.


*I had multiple partners, now I stick with one person. I am even living a healthier lifestyle, eating healthy too (Mother of a 6-year-old).*



*It has changed me because of the partners I had been with, it made me more responsible. I was not using condom before but now I use it. We also changed our diet (Mother of a 9-year-old).*



*My life has changed in many ways. I use to drink, but now I have limits in all things I do about my life (Mother of a 10-year-old).*


Motherhood was also frequently mentioned as a motivator for engaging in positive coping mechanisms and adhering to treatment.


*I am no longer self-centred, having a positive child has thought me to care about others and not just myself. When I had no child, I would move around freely but now I restrict my movements (Mother of an 8-year-old).*



*I am HIV positive and my child is HIV positive, therefore, I have to look after myself also. I want to live for my children and to look after them (Mother of an 8-year-old).*


## 4. Discussion

This study explored the meaning of an HIV diagnosis during pregnancy for women in public facilities in a South African health district. The findings showed that getting HIV-positive results during pregnancy awakened intense feelings of fear. At the time of testing positive, the women experienced high emotional intensity, emotions that were similar to those reported in other studies [17,26,40]. The diagnosis of HIV was a traumatic and life-altering event, and they thought that death was imminent shortly after diagnosis. Barkish et al. [41] argue that the progress made in the field of ART has not been successful in changing people’s perceptions about an HIV positive diagnosis. The perception and conceptualisation of HIV diagnosis as death, persist in many societies [17,42]. For the women in the study, the diagnosis of HIV coincided with pregnancy, transforming pregnancy from a period of excitement to one of anxiety [25]. The transition from being HIV negative to becoming HIV positive and pregnant was overwhelming for the women. This phenomenon was reported elsewhere [3]. Kelly et al. [17] refer to the timing of the diagnosis of HIV as the centre of the disruption of the lives of the women in their study.

The accounts of the women revealed that they felt as though life had ended after an HIV diagnosis. Previous studies reported similar traumatic emotions [16,17,40]. Nurse clinicians should be sensitive to the unique set of emotional challenges that women diagnosed with HIV during pregnancy experience [26]. Of note is that the reactions observed are despite therapeutic advances in the ART that have redefined HIV as a chronic disease and echoed findings in other studies [23,40,41]. The women viewed HIV as a fatal illness despite all of them being asymptomatic at the time of receiving positive test results and had access to ART through PMTCT services. Baumgartner and Keegan [43] found that although women in their study knew that ART medications existed, fear of imminent death was common. In South Africa where the burden of HIV is high and HIV testing in pregnant women is routine, the issues related to an HIV diagnosis continue to affect the lives of women and their families in a negative way [17]. Educating women on the effectiveness of ART will reduce trauma and mitigate the impact of the positive test results on their wellbeing [14,15].

The findings of this study also highlighted that the women struggled to accept the HIV diagnosis and the consequences that it had for their lives and that of their unborn children. These findings corroborate those from other studies [14,15,16]. Denial came about because most of the women did not perceive themselves to be at high risk for contracting HIV infection. Research suggests that the denial of risk prior to the diagnosis is an indication that HIV had not been a concern for the women, because HIV happen to other people [3,17]. Most of the women in the study had only one partner at the time and they related HIV infection with promiscuous behaviour. Research noted that women perceive having one sexual partner as a protective factor against becoming HIV positive [15,16]. In contrast, those who questioned their morality, such as engaging in unsafe sexual behaviours, reported feelings of guilt, self-blame, and shame, which contributed to non-acceptance.

The immediate impact of an HIV diagnosis was severe, leading to feelings of depression and reduced self-worth. Other researchers reported similar observations [14,23,26]. Women were emotionally unprepared for an HIV diagnosis, were in denial, and were vulnerable to depression after their diagnosis. They reported self-imposed isolation or withdrawal from social activities as well as disengaging from regular activities. Some of them reported drastic changes in their lives such as self-neglect. Several studies identified denial, emotional distress, and isolation as initial responses to an HIV diagnosis [14,16,26]. However, for the women in this study, the depression that followed the diagnosis of HIV persisted for months or years. They reported perpetual feelings of guilt, self-blame, hopelessness, and constant anxiety. Ashaba and colleagues [44] are of the view that the psychosocial challenges HIV positive pregnant and postpartum women face, are not addressed in HIV or antenatal clinics. Incorporating psychosocial care within the PMTCT programme will facilitate acceptance of self and HIV diagnosis [26].

Being pregnant and testing HIV positive also triggered anxious concerns and excess worries about their unborn child, despite medical reassurance and being initiated on or offered ART. This became a source of great emotional distress for them. Other studies reported similar concerns [3,23,24,25,26]. During pregnancy, the fear that the child might be infected preoccupied the women, and upon a positive diagnosis of their child, they lamented over this and were inconsolable. Prior to confirmation of the child’s HIV diagnosis, the maternal levels of anxiety are significantly increased [25,45,46]. The findings underscore the significance of providing timely psychosocial support to women post-diagnosis as part of the cascade of care in PMTCT to help them overcome fears of vertical transmission and engage in HIV care [25,44].

Upon a positive HIV diagnosis of their children, the pain the mothers experienced was greater than the pain they felt when they learned of their own HIV diagnosis. Similar to the reaction to their own diagnosis, the mothers were burdened by constant thoughts of death, and they feared that their children might die prematurely. Raising a child with HIV occurred within a context of high stigma and discrimination. Thus, secrecy emerged as one of the ways the mothers used to protect their children from stigma and discrimination. Although the mothers believed that secrecy was in the best interest of their children, the current study and others found that the mothers had limited sources of support as secrecy of the HIV diagnosis of the child limited their potential support network to listen to their pain and challenges [47,48,49].

Acceptance of own HIV diagnosis in the current study was non-linear and ranged from no acceptance at all to full acceptance. The acceptance echoed patterns from previous studies [13,50]. Although in the beginning there was slow progress towards acceptance, with the passage of time, the women transitioned to feelings of acceptance. The findings are in agreement with prior studies [13,26]. For example, Contreras et al. [26] indicated that the majority of women in their study ultimately achieved emotional stability. Many women in the current study gained self-acceptance due to having disclosed their own HIV diagnosis and that of their children. For most participants, the outcome of disclosure provided opportunities to receive social support, especially from family. In addition, they received ART as part of the cascade of care provided through PMTCT services and when they did not get ill at all, they began to accept their HIV diagnosis. However, the finding resonates with those from other studies that some women experienced non-acceptance for extended periods of time [13,40]. Similar findings were reported among heterosexual adults living with HIV. Kutnick and colleagues [47] reported that for some individuals, years passed while they struggled to become ready to accept their HIV diagnosis. The findings highlight the importance of providing effective and timely psychological, social, and practical support to women following a diagnosis of HIV during pregnancy [25].

Prolonged denial or non-acceptance of HIV diagnosis has serious potential adverse public health consequences for the individual, such as delayed engagement in HIV care and uptake of ART [13,14,50]. Given that the focus of the PMTCT programme is to initiate ART at the point of testing to eliminate vertical transmission of HIV, delayed engagement in care for women with an HIV diagnosis increases the risk of vertical transmission. Therefore, integrating continuous HIV counselling into PMTCT practice or the HIV care continuum could foster acceptance for pregnant women with HIV diagnosis [14]. Furthermore, encouraging participation in formal and informal support groups has been identified as a powerful means to increase the emotional well-being of women living with HIV [26]. For the women in this study, participating in support groups gave them purpose, motivation to engage in positive coping, and a sense of empowerment.

Denial is a psychological defence strategy that women used to cope with distressing feelings of the HIV diagnosis that compromised early engagement in HIV-related care. The acceptance of self and acceptance of HIV diagnosis was a key coping strategy used by women. The findings are in agreement with previous studies [44,51]. Acceptance was a necessary step towards overcoming denial and engaging in HIV care in the current study as in others [44,50]. Other researchers have found that acceptance came with a growth in personal resilience, which empowered women to cope with the challenges of a life with HIV [16]. The findings showed that survival was a strong motivator for engaging in self-care, and motherhood increased their determination to survive. Similar findings were reported elsewhere [14,17,25,46,52]. The accounts of the women revealed that they engaged in self-management strategies to make positive lifestyle changes. They had trust in the efficacy of the ART regimen and adhered to treatment in order to survive and raise their children [15,52].

### Study Limitations

The study had some limitations, like most qualitative studies that included a small study sample that may limit generalisability of the study findings to all women who receive a positive HIV diagnosis during pregnancy. The experiences of women and the recovery process in this study may or may not be similar to other women who test positive in pregnancy and the postpartum period in other settings. Secondly, the time since diagnosis for some of the women was recent, whereas for others a longer period had elapsed which might have affected the recall for the mothers. Recalling the time it took them to accept their status might be affected by the time since diagnosis. On the other hand, the varied period may have facilitated a more reflective account of the experiences of the women and the process to recovery.

## 5. Conclusions

The present study extends the literature on how women respond to an HIV positive diagnosis in pregnancy and the process of acceptance of the diagnosis and self. Testing HIV positive during routine screening and being free of symptoms increased the shock, disbelief, and strong emotions exhibited in the study.

This study’s accounts resonated with other research that acceptance of HIV diagnosis is essential for engagement with HIV care. Albeit the process of acceptance was varied and contextual, the women eventually changed their perceptions of HIV and accepted the diagnosis as a lifelong chronic illness. The transition towards acceptance of HIV diagnosis and motherhood was influenced by various factors including disclosure to significant others, receiving support, the need to survive, trust in ART, and hope for life after diagnosis.

Nurse clinicians are the first point of contact for pregnant women diagnosed with HIV during their participation in the PMTCT programme. It is therefore imperative that they are appropriately trained to recognise the significance the women attach to the HIV diagnosis. Having a deep understanding of the meaning of an HIV diagnosis is essential for providing the psychosocial care needed by the women to minimise the negative impact of HIV positive test results. This could be achieved by developing ongoing counselling and support interventions outside the post-test counselling offered to women enrolled in PMTCT. In agreement with other recommendations, there is need for counselling over multiple sessions to allow time for the women to process, accept the HIV diagnosis, and engage in HIV care. The findings further underscore the need to incorporate culturally appropriate psychosocial care and mental health services in PMTCT services. In addition, nurses should provide information on support groups following HIV diagnosis and repeat this information throughout pregnancy, post-delivery, and the HIV care continuum to foster acceptance of the HIV diagnosis.

## Figures and Tables

**Table 1 ijerph-18-13006-t001:** Summary of themes.

Themes	Sub-Themes
The unanticipated HIV diagnosis	
Fear of premature death	
Extreme emotional distress	Despair
	Guilt, shame, and blame
	Worrying about the unborn baby
	Self-neglect and depression
Disclosure of HIV diagnosis	Delayed disclosure
	Outcome of disclosure
Acceptance of self and HIV status over time	Acceptance is a process
	Counselling gives hope
	Trust in ART
	The importance of support from friends and families
Life changes after acceptance	

## Data Availability

All the data is included in the analysis of the study.

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
