# Peer review of "When Pregnancy Coincides with Positive Diagnosis of HIV: Accounts of the Process of Acceptance of Self and Motherhood among Women in South Africa"

_ijerph, 2021, doi:10.3390/ijerph182413006_

Round 1
Reviewer 1 Report
Please consider providing more detailed comments based on the questions
below:
1. What is the main question addressed by the research?
The main question addressed in this paper is the exploration of the experiences of women who tested positive for HIV during pregnancy
2. Do you consider the topic original or relevant in the field, and if so, why?
Although HIV has been with us for a long time now, there still areas that have not been adequately explored. This includes the emotional experiences of women who have to deal with a diagnosis of a condition that still carries stigma on top of having to deal with a pregnancy, the fear of vertical transmission to the infant and the process of recovery.
3. What does it add to the subject area compared with other published
material?
A lot of studies on HIV/AIDS in pregnancy have addressed the numbers (Prevalence) and associated factors. While knowing the prevalence of say- depression or anxiety is important, very few have looked at the actual experience of the women. Qualitative studies have this and the added benefit of allowing the voices of the participants to be heard through the narrative. This study highlights the need, for continued support of the women diagnosed with HIV during pregnancy, far beyond administering ARVs. The voices of this women come across in a way that numbers and statistics would never come across. Making this paper readable and understandable by health providers and carers who are not academicians.
4. What specific improvements could the authors consider regarding the
methodology?
One limitation in this paper is that the interviews were done, sometimes, years after the pregnancy. Stories change with time and are coloured by the present. All the participants interviewed had the added stress of having delivered and living with children who are HIV positive. This is an aspect of the study that the reader is not expecting given that it is not mentioned in the research question. It would have added more either to expand the research question, to cover this aspect or if the selection had also included women from the same cohort who had delivered HIV negative babies.
5. Are the conclusions consistent with the evidence and arguments presented and do they address the main question posed?
The conclusions are consistent with the evidence based on the narrative of the women.
- Are the references appropriate?
The references are comprehensive and appropriate
7. Please include any additional comments on the tables and figures.
Not applicable given that this is a qualitative study
Author Response
I would like to thank the reviewers for their positive comments, there were no suggestions or recommendations to be addressed.
Reviewer 2 Report
Thank you for this very interesting and important paper that I enjoyed reading. It reports from an important study exploring the experience of women receiving a positive HIV diagnosis during pregnancy. It’s also pleasing to see the findings of from a qualitative study, seeking the lived experience of women in this challenging period.
The paper is written well and is generally structured appropriately for reporting from this type of study, and I have only a few minor suggestions for improvement.
INTRODUCTION
This provides an important background to the topic and justification for the study. On line 84, this this be ‘few’ studies have’ rather than ‘a few’ studies?
MATERIALS AND METHODS
The design and analysis are well-described and suitable for a qualitative study.
I note that the study has ethical approval, and the consenting process appears acceptable. I also note the measures taken to protect privacy and confidentiality of the participants.
RESULTS
The results provide useful insights into the lived experiences of women in this context. Many of these of course are typical of many women receiving a positive HIV diagnosis, pregnant or not, and it’s useful to see the additional factors here relating to fear for the health of their unborn children. The support of friends and family is important, and it’s useful that you’ve highlighted this component. I wondered here if it was all positive? There is likely to have been some degree of stigma experienced for some, especially from other people feeling overprotective of the unborn child. If there are more data on this it would be interesting to see.
The use of quotations is effective.
DISCUSSION/CONCLUSION
This section draws appropriate on the data and provides additional citations to place the findings in what is currently known. Many of your findings are similar to other studies, but you do highlight some of the particular factors revealed in your data. You make the important point that acceptance of an HIV diagnosis is not linear, with negative emotions circulating for some time.
One other point of interest – I couldn’t see much on how a mother’s recall of her experience was impacted (or not) by the child going on to develop HIV, except in one case. I would think this would have a significant impact on a mother’s memory of the pregnancy/HIV diagnosis. I presume for the women receiving the diagnosis during pregnancy in this study the child would not go on to develop HIV. Is this the case? Your sample selection criteria did include at least one child living with HIV so was this from a previous pregnancy? There could be more clarity here, and, if data are available, to expand this point it would be interesting to widen the discussion a little.
One point on writing style – some paragraphs in your discussion begin with statements that may or may not relate directly to your study. It may be useful sometimes to start a paragraph with ‘In our study’ or ‘We also found that’ or similar.
You include appropriate limitations for qualitative studies, and that recall may be different (some of the mothers were diagnosed some years before). Your recommendations are appropriate – and perhaps the point that support and care should be long-term should be made more strongly.
Reviewer recommendations
- Minor typos, and consider line 84
- Suggestion: If data are available about stigma from family or friends this could be included (the findings suggest they were all supportive – is this the case?)
- Provide a little more clarity about the impact on the mother of a child going on to develop HIV. Were the HIV positive children of the participants in this study the result of previous pregnancies where the mothers were undiagnosed, for example, or of the same pregnancy as when they received the diagnosis. This would affect recall and experiences.
- Suggestion: Add text in discussion section to clarify the point is based on your study (‘We also found that’ or something).
- Suggestion: Strengthen the recommendation for long-term support and care (given that, as you found, negative emotions can be especially resilient)
Author Response
I would like to thank the reviewers for their positive comments. All the comments and suggestions from the reviewer have been addressed and highlighted blue in the relevant sections of the manuscript. In addition, I have highlighted the responses below.
- Minor typos, and consider line 84-corrected
- Suggestion: If data are available about stigma from family or friends this could be included (the findings suggest they were all supportive – is this the case?) Response-I added a theme on disclosure of the diagnosis of HIV (table 1) to address the issue of delayed disclosure, stigma, and discrimination. The quotes also show how the mothers used secrecy to prevent stigma (from line 331). Line 531-537 in the discussion, I present arguments about stigma and discrimination and the effects of secrecy on receiving social support.
- Provide a little more clarity about the impact on the mother of a child going on to develop HIV. Were the HIV positive children of the participants in this study the result of previous pregnancies where the mothers were undiagnosed, for example, or of the same pregnancy as when they received the diagnosis? This would affect recall and experiences. Response-I added a narrative about the mothers’ reactions to learning about the HIV diagnosis of their children in the results section (line 262-270). In the discussion, line 528-531, the impact of the HIV diagnosis of the child on the mother is discussed.
- Suggestion: Add text in discussion section to clarify the point is based on your study (‘We also found that’ or something). Response-the suggestions is addressed in paragraphs 2, 4, 7, and 8.
- Suggestion: Strengthen the recommendation for long-term support and care (given that, as you found, negative emotions can be especially resilient). Response- I added an additional sentence to strengthen the recommendations.